# Cohort-profile: Household transmission of SARS-CoV-2 in a low-resource community in Rio de Janeiro, Brazil

Patrícia Brasil [1], Luana Damasceno,[1] Trevon Fuller,[1,2] Leonardo S Bastos,[3] Oswaldo G Cruz,[3] Fernando Medeiros,[1] Guilherme Amaral Calvet [4], Paola Resende,[5] Jimmy Whitworth,[6] Chris Smith [7,8] Marilda M Siqueira,[9] Marilia Carvalho[3]

**Correspondence to**
Dr Patrícia Brasil;
patricia.brasil@ini.fiocruz.br

## ABSTRACT

**Purpose** To better understand the household transmission of SARS-COV-2 in a low-resource community in Rio de Janeiro during the COVID-19 pandemic (2020–2022).

**Participants** This is an open prospective cohort study of children ≤12 years old and their household contacts. During home visits over 24 months, we collected data on sociodemographic characteristics, behavioural data, clinical manifestations of SARS-CoV-2, vaccination status, SARS-CoV-2 (reverse transcription-polymerase chain reaction) RT-PCR and anti-S antibody tests. Among adults, the majority of participants were women (62%).

**Findings to date** We enrolled 845 families from May 2020 to May 2022. The median number of residents per household was four. The median household density, defined as the number of persons per room, was 0.95. The risk of SARS-CoV-2 occurrence was higher in households with a high number of persons per room. Children were not the principal source of SARS-CoV-2 infections in their households during the first wave of the pandemic.

**Future plans** Future studies will investigate cellular and humoral immune responses to locally circulating SARS-CoV-2 variants, which is relevant for the design of vaccines, antivirals and monoclonal antibodies. We will also engage in outreach to encourage vaccination as a means of limiting the transmission of novel SARS-CoV-2 variants and other emerging pathogens.

## INTRODUCTION

The emergence of COVID-19 has caused a serious health crisis affecting the world since 2020. More than 624 million confirmed cases, including 6.6 million deaths, have been reported up to October 2022 worldwide.[1] In the Western Hemisphere, more than 20% of all COVID-19 cases have occurred in Brazil. At the global scale, 10% of COVID-19 deaths have been in Brazil. Rio de Janeiro has reported the second highest number of COVID-19 deaths of any city in Brazil.[2 3]

Understanding intrafamily transmission is essential for designing appropriate interventions. Much of the research investigating this question has been based on cross-sectional

### STRENGTHS AND LIMITATIONS OF THIS STUDY

⇒ Rates of participant retention and visit completion were high when compared with similar settings.
⇒ We were able to estimate the overall incidence of SARS-CoV-2 infection in this population, not restricted to individuals with obvious clinical signs of COVID-19.
⇒ Since the cohort has been followed continuously for 2 years, we have been able to gather data about successive variants that have swept through the population.
⇒ There was some irregularity of study visits, which sometimes had to be rescheduled due to armed conflicts in the community.
⇒ We could have missed acute SARS-CoV-2 infections either between visits or when visits were cancelled.

studies that do not accompany participants longitudinally making it difficult to investigate the transmission process over time.[4] Longitudinal data about household transmission are particularly scant in low-income and middle-income countries. Among the factors that may drive such transmission are the difficulty of remaining isolated in crowded households and insufficient access to laboratory tests. In addition, transmission patterns have evolved in the 2 years of the pandemic with circulation of SARS-CoV-2 variants, which differ in virulence and transmissibility.

To better understand the local transmission of SARS-COV-2 in the aforementioned communities and the role of children in such transmission, we have followed a cohort of children and their household contacts during the COVID-19 pandemic period (2020–2022).

## COHORT DESCRIPTION

This is an open prospective cohort study of children ≤12 years old and their household contacts. The present report describes our

findings from 20 May 2020 to 31 May 2022. Recruitment was closed at the end of this analysis, but it is immediately reopened whenever a new variant of concern is detected in the community.

## The epidemic context

This period encompassed different locally-circulating SARS-CoV-2 lineages, in particular, B.1.1.33 during the first wave[5] followed by variants Zeta (P.2), Gamma (P.1/P.1.*), Delta (B.1.617.2/AY.*) and Omicron (BA.*). The epidemiological scenario in the community has evolved during the study.[6] In March 2020, social distancing policies including quarantine and school closure were implemented. From April 2020 to October 2021, emergency financial assistance was made available to low-income households. In June 2020, bars, stores and restaurants were reopened. However, adherence to control measures varied among neighbourhoods and was generally poorest in low-resource communities. Vaccination began in Rio de Janeiro in January 2021. Public schools resumed in-person classes in November 2021, immediately before the end of the academic year.

## Recruitment and eligibility criteria

Recruitment took place at the Germano Sinval Faria Health Centre, at the Sergio Arouca National School of Public Health, Oswaldo Cruz Foundation. This primary healthcare centre is located in Manguinhos, a neighbourhood in the Northern sector of the city of Rio de Janeiro. It provides primary care services to residents of Manguinhos, which is a neighbourhood with low household income. The clinic provides medical care free of charge to people who live in Manguinhos serving an average of 30000 adults and 4300 children≤12 years old every month.

All children brought to the primary local health centre for immunisation, check-ups or emergency care were eligible to participate in the study, irrespective of whether or not they were symptomatic. At this time, the accompanying adult was invited to participate. On enrolment, a study nurse administered a questionnaire to the child's parent or guardian to collect data about the child's sociodemographic and clinical characteristics and medical history, as well as characteristics of the household. The questionnaire was completed on a tablet and took approximately 20 min. If the child was symptomatic, an additional clinical questionnaire was completed by a paediatrician requiring approximately 30 more min. At the same time, we collected nasopharyngeal and serum samples from the child and any accompanying person. We also obtained the street address of the child's home, the parent or guardian's workplace (if different) and the telephone numbers of any relatives. This information was used to schedule home visits. Those who were approached but did not enrol in the study were either adults who stated that they did not have time to participate or children whose parent or legal guardian was not present at the time of the visit and therefore consent could not be obtained. A translation of the informed consent form is included in online supplemental material 1.

## Follow-up

In each household, we collect samples on days 1 (first visit), 14 and 28, every 3 months during the first year and twice in the second year. All symptomatic participants were evaluated and provided with medical care until symptoms resolved. The total follow-up time for each family was 24 months. At households with any symptomatic resident, an additional sample was collected on day 7. Households in which no individual was examined during the entirety of a given epidemic wave were considered lost to follow-up after eight unsuccessful contact attempts by telephone and smartphone messaging apps. Individuals in each household who missed one or more study visits were excluded only if the study team was unable to contact the participant during the entire wave. Individuals with only one study visit were considered lost to follow-up.

Home visits were scheduled by phone and conducted by a team of field workers, including paediatricians and nurses. During these visits, study participation was offered to the child's household contacts, nasopharyngeal and serum samples were collected, and the Global Positioning System (GPS) coordinates of the household were recorded. The study personnel completed a questionnaire for each contact who consented to participate consisting of clinical history, sociodemographic characteristics and vaccination status. Visits lasted an average of 40 min. If time constraints prevented a participant from responding to all of the questions during the visit, the study team followed up with him or her by phone to complete the questionnaire.

## Data collected

The structure of the participant database is shown in figure 1. The information collected at baseline and each subsequent visit is summarised below:

► Sociodemographic characteristics: age, education, race, profession; residence: address, GPS coordinates, telephone, number of people living in the household and number of rooms. With respect to race, the only legally accepted form of racial identification in Brazil is self-declaration. Data on race, schooling and smoking were only collected for people 18 years of age (yoa) and older.

► Behavioural data: mask use, use of public transportation, visits to restaurants, bars, malls, movie theatres, barbeques, birthday parties or other events where there was a crowd.

► Clinical manifestations in children: seizures, anosmia, dysgeusia, food refusal, fever, cough, nasal congestion, prostration, shortness of breath, abdominal pain and diarrhoea.

► Adolescent/adult clinical variables: anosmia, dysgeusia, loss of appetite, fever, cough, rhinorrhoea, headache, prostration, shortness of breath, sore throat, abdominal pain and diarrhoea.

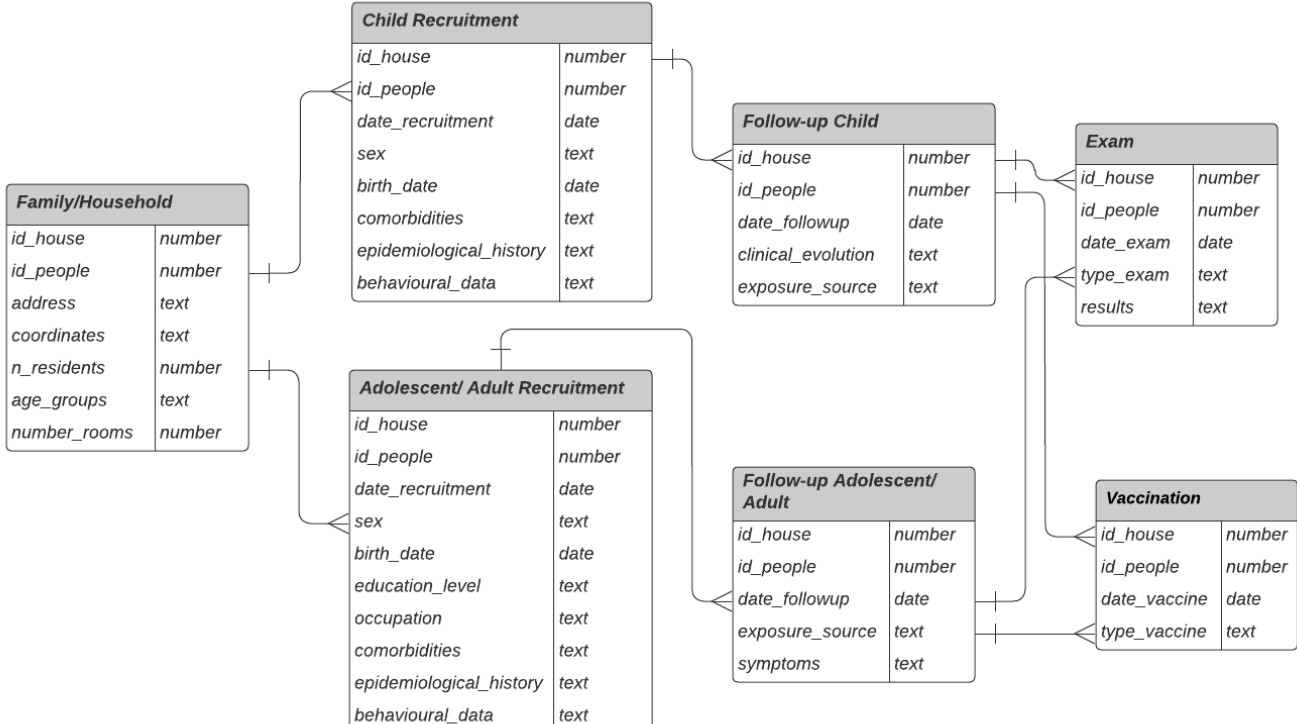

**Figure 1** Cohort database structure.

► Vaccination status: number of doses, manufacturer and vaccination date (after vaccination began in January 2021).
► Laboratory test results: SARS-CoV-2 RT-PCR test, SARS-CoV-2 lineage genomic test and SARS-CoV-2 IgG anti-S antibody test of serum samples.

Nasopharyngeal swabs were tested by real time RT-PCR to amplify the E gene and the RdRp region of the Orf1ab gene of SARS-CoV-2 (Charité/Berlin, Germany). Cycle thresholds (CT) less than 40 were classified as positive. We assembled SARS-CoV-2 genomes from Illumina amplicons using the CLC Genomics Workbench V.20.0.4 (QIAGEN). Phylogenetic relationships among SARS-CoV-2 samples were determined using the Pango Lineages tool. SARS-CoV-2 serology (IgG) was performed using a Microparticle Chemiluminescent Immunoassay, targeting the S gene (Abbott Laboratories, Abbott Park, Illinois, USA). All assays were performed according to the manufacturer's instructions. A study database was created with REDCap.[7 8]

## FINDINGS TO DATE

In total, 79.5% of the individuals living in the same household as the child when approached consented to participate in the study. A total of 845 families were enrolled in the study: 34.2% in the first epidemic wave, 16.4% in the Zeta wave, and 23.9% in the Gamma wave, 18.1% in the Delta wave, and 7.4% in the Omicron wave. This included 1152 children 12 and under, 758 of whom were retained in follow-up. As quarantine measures were relaxed, parents returned to working outside the home, which led to loss to follow-up of adults. This also resulted in loss to follow-up of children because children could not receive the study team in their homes without an adult present A total of 7068 RT-PCR and 5432 serology tests were performed.

We collected just one sample from 39.4% of the participants, and therefore, they were considered lost to follow-up. The racial composition of the lost to follow-up group and the retained in follow-up group were similar, as were years of schooling. However, the percentage of participants who were women differed between the groups. In the lost to follow-up group, 58% were women, whereas 62% of participants retained in follow-up were women (Table 1). As the individuals eligible to participate in the study came from households with more women than men, it is unsurprising that women were more likely to be present during the study visits and be retained in follow-up than men. The retained and lost to follow-up groups also differed with respect to the proportion who were adults 18 years of age or older. Fifty-eight per cent of the lost to follow-up group were 18 or older vs 48% in the retained in follow-up group.

The average number of participants recruited per week was 27.1. The period of follow-up was 24 months. As enrolment began in May 2020 and we followed participants through May 2022, some participants had completed their schedule of visits before the end of the

**Table 1** Demographic characteristics and schooling of individuals lost to follow-up versus those retained in the cohort

| Variables | Categories | Lost to follow-up | Retained in follow-up | P value |
|---|---|---|---|---|
| Age (years) | 0–12 | 404/1143 (35%) | 758/1760 (43%) | <0.001 |
| | 13–17 | 74/1143 (7%) | 153/1760 (9%) | |
| | 18 or older | 665/1143 (58%) | 849/1760 (48%) | |
| Sex | Female | 658/1143 (58%) | 1095/1760 (62%) | 0.012 |
| | Male | 485/1143 (42%) | 665/1760 (38%) | |
| Race* | Non-white | 433/665 (65%) | 571/849 (67%) | 0.7 |
| | White | 197/665 (30%) | 250/849 (30%) | |
| | Missing | 35/665 (5%) | 28/849 (3%) | |
| Completed primary school* | Yes | 473/665 (71%) | 651/849 (77%) | 0.2 |
| | No | 138/665 (21%) | 159/849 (19%) | |
| | Missing | 54/665 (8%) | 39/849 (4%) | |

*Race and schooling were only collected for adults.

study (figure 2). Those who were recruited recently have not completed all of their follow-up visits. Currently, 67.5% of the participants who were enrolled at any time are still being followed. The mean follow-up time was 235 days for participants ≥13 years old and 261 days for those ≤12 years old. The mean follow-up time was greater for women ≥13 years old (247 days) than for men in the same age group (208 days).

Our cohort can be analysed at the household scale or at the individual scale. At the household scale, the sociodemographic composition of the domiciles was similar throughout the study. The median number of residents per household was four. The median household density, defined as the number of persons per room, was 0.95. The cohort can also be analysed at the individual scale, without considering the household to which each individual belonged. At the individual scale, most sociodemographic characteristics were similar across the entire period, except for the Omicron wave, in which there were fewer participants (table 2). The majority of participants were women (62.3%), reflecting not only the recruitment strategy, as children are usually accompanied by their mothers or grandmothers, but also the caregiver at home. The median age of the participants was 14–19 during the first four waves of the participants but decreased to 9 years of age during the Omicron wave. Across all five waves of the pandemic, the IQR of participant ages was approximately 5–35 years old. The self-declared race of 70% of the adult participants was non-white. Across all recruitment periods, 45%–54% of the participants were adults, and 36%–52% were children ≤12 years old (table 2). More than 80% of the adult participants had completed primary school. The prevalence of smoking among adults was 9%.

The seroprevalence of SARS-CoV-2 antibodies indicative of past exposure to the virus or a vaccine was approximately

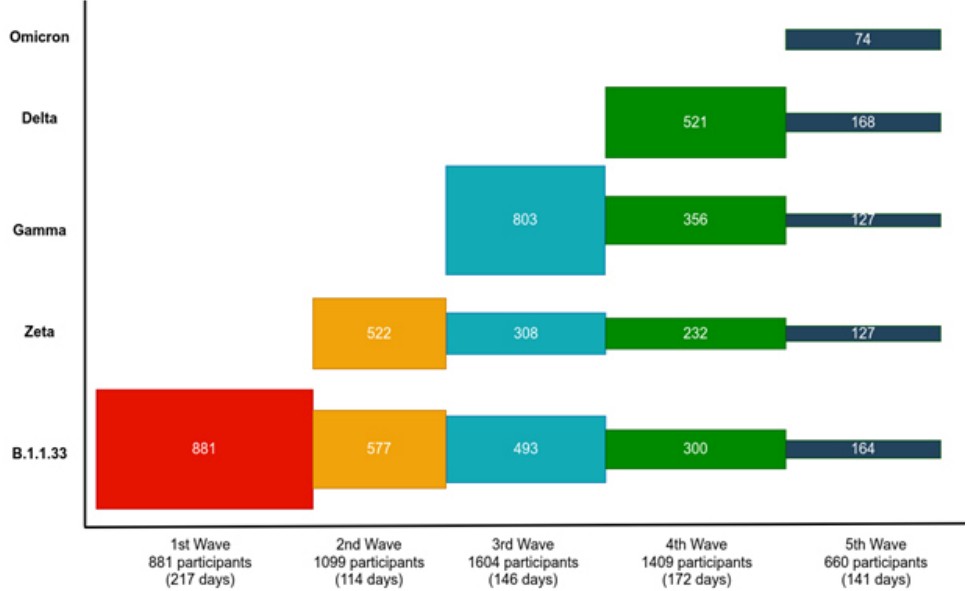

**Figure 2** Recruitment and retention by SARS-CoV-2 epidemic wave.

**Table 2** Demographic characteristics of the cohort participants by epidemic wave

| Variable | Overall Median (IQR) | B.1.1.33 Median (IQR) | Zeta Median (IQR) | Gamma Median (IQR) | Delta Median (IQR) | Omicron Median (IQR) |
|---|---|---|---|---|---|---|
| Days of follow-up | 222 (105–365) | 317 (196–522) | 265 (147–426) | 162 (68–256) | 137 (49–197) | 27 (14–93) |
| Age (years) | 16 (7–35) | 14 (5–36) | 19 (7–35) | 19 (8–38) | 15 (8–32) | 8 (4-24) |
| | N (%) | N (%) | N (%) | N (%) | N (%) | N (%) |
| **Sex** | | | | | | |
| Male | 665/1760 (38) | 239/623 (38) | 131/343 (38) | 174/464 (37) | 106/287 (37) | 15/43 (35) |
| Female | 1095/1760 (62) | 384/623 (62) | 212/343 (62) | 290/464 (63) | 181/287 (63) | 28/43 (65) |
| **Age groups** | | | | | | |
| 0–12 | 758/1760 (43) | 289/623 (46) | 138/343 (40) | 173/464 (37) | 129/287 (45) | 29/43 (67) |
| 13–17 | 153/1760 (9) | 46/623 (8) | 28/343 (8) | 51/464 (11) | 28/287 (10) | 0 |
| >=18 | 849/1760 (48) | 288/623 (46) | 177/343 (52) | 240/464 (52) | 130/287 (45) | 14/43 (33) |
| **Race*** | | | | | | |
| White | 250/849 (29) | 81/288 (28) | 57/177 (32) | 71/240 (30) | 36/130 (28) | 5/14 (36) |
| Non-white | 571/849 (67) | 204/288 (71) | 113/177 (64) | 154/240 (64) | 91/130 (70) | 9/14 (64) |
| Missing | 28/849 (4) | 3/288 (1) | 7/177 (4) | 15/240 (6) | 3/130 (2) | 0 |
| **Completed primary school*** | | | | | | |
| Yes | 651/849 (77) | 219/288 (76) | 138/177 (78) | 179/240 (75) | 105/130 (81) | 10/14 (71) |
| No | 159/849 (19) | 65/288 (23) | 28/177 (16) | 43/240 (18) | 19/130 (15) | 4/14 (29) |
| Missing | 39/849 (4) | 4/288 (1) | 11/177 (6) | 18/240 (7) | 6/130 (4) | 0 |
| **Smoking*** | | | | | | |
| Yes | 73/849 (9) | 28/288 (9) | 10/177 (6) | 11/240 (5) | 16/130 (13) | 8/14 (57) |
| No | 752/849 (89) | 258/288 (90) | 160/177 (90) | 217/240 (90) | 111/130 (85) | 6/14 (43) |
| Missing | 24/849 (2) | 3/288 (1) | 7/177 (4) | 11/240 (5) | 3/130 (2) | 0 |

*Race, schooling and smoking were only collected for adults.

20% from May 2020 to June 2021. Seroprevalence subsequently increased precipitously, likely reflecting the rollout of SARS-CoV-2 vaccination (figure 3). Complete vaccination (two doses of BNT162b2, AZD1222, CoronaVac (Sinovac) or one dose of Ad26.COV2.S) increased with age and was over 80% for all but the youngest age groups (figure 4).

The first published report from this cohort provided evidence that children were not the principal source of SARS-CoV-2 infections in their households during the first wave of the pandemic but more frequently acquired the virus from adolescent and adult household contacts.[9] In a subsequent study, we estimated the incidence of SARS-CoV-2 in the cohort during the B.1.1.33, Zeta, Gamma and Delta waves.[6] Household crowding was significantly associated with SARS-CoV-2 incidence only during the first wave.[6] Through the Delta wave, the incidence among children was lower than that of older participants. Subsequently,

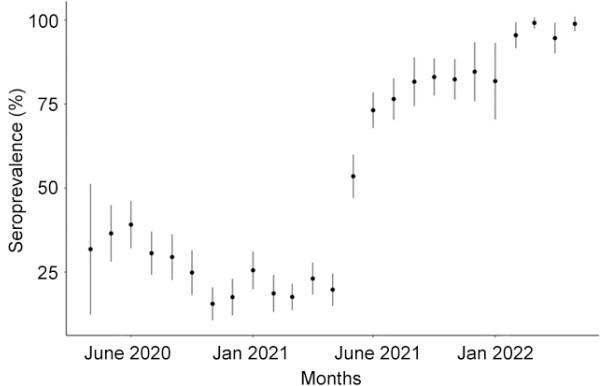

**Figure 3** SARS-CoV-2 anti-S seroprevalence by month of sample collection.

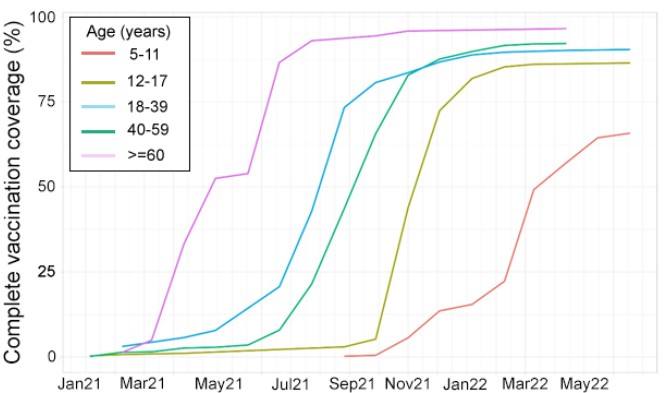

**Figure 4** SARS-CoV-2 vaccine coverage by age group.

vaccination of the elderly reached 90% and incidence among children was higher.

## Patient and public involvement

The study personnel include residents of Manguinhos who were first hired during our previous cohort studies.[9–13] These professionals are considered trustworthy by members of the community and permitted to move through the area freely. This enables them to find households that would be extremely difficult for someone unfamiliar with the community to locate. In addition, all study participants were given the opportunity to provide feedback about the design of the study during the first home visit.

## STRENGTHS AND LIMITATIONS OF THIS STUDY

Our study data are unique because this is a community that is not easily accessible to researchers due to high levels of poverty and violence. Nevertheless, it is emblematic of low-resource communities in other cities in Latin America. The study's high rates of participant retention and visit completion are among its principal strengths. Furthermore, since we recruited asymptomatic and symptomatic participants, we were able to estimate the overall incidence of SARS-CoV-2 infection in this population, not restricted to individuals with obvious clinical signs of COVID-19. Our study team included personnel from different disciplines including epidemiology, medicine, virology, immunology and statistics. We believe that this strengthened the research because we were able to examine the ramifications of the COVID-19 pandemic for this community from a number of different angles. As the field workers were people living in the community, they were able to easily establish a link with participants. Finally, since the cohort has been followed continuously for 2 years, we have been able to gather data about successive variants that have swept through the population. The main limitation is that in some cases it was not possible to carry out visits in accordance with the timeline in the study protocol. The principal reasons for missed visits were armed conflicts within the community or work obligations outside the community. We may have failed to detect acute SARS-CoV-2 infections that occurred at the time of missed visits.

## COLLABORATION

While the data at the individual level are not openly available to preserve the anonymity of the study participants, researchers who are interested in potential collaboration should contact the corresponding author. The study steering committee will evaluate data requests in accordance with best practices in cohort studies to ensure that data sharing would preserve confidentiality and be compatible with the terms of the informed consent form used in this study.

## FUTURE PLANS

The biobank of cryopreserved peripheral blood mononuclear cells and serum from the cohort participants will allow future studies of cellular and humoral immune responses to locally circulating SARS-CoV-2 variants, which is relevant for the design of future vaccines, antivirals and monoclonal antibodies. We will also analyse the immune profile of participants who experienced breakthrough infections following vaccination. Of course, in order for novel SARS-CoV-2 variants and other emerging pathogens to be controlled, it will be necessary to reduce transmission. To this end, future work will also involve publicising virus control and prevention practices. We thus plan to create a short 'edutainment' style video to disseminate our results and encourage SARS-CoV-2 immunisation. The target audience will be children and families. As part of these efforts, we will work closely with media partners, policymakers and the cohort participants to develop effective outreach strategies. We believe such activities might help bring about better compliance with COVID-19 control measures not only in this community, but also at the regional and global levels.

### Author affiliations

[1]Instituto Nacional de Infectologia Evandro Chagas, Fundacao Oswaldo Cruz, Rio de Janeiro, Brazil
[2]Institute of the Environment & Sustainability, University of California Los Angeles, Los Angeles, California, USA
[3]Scientific Computing Program, Fundacao Oswaldo Cruz, Rio de Janeiro, Brazil
[4]Acute Febrile Illnesses Clinical Research Laboratory, Oswaldo Cruz Foundation, Rio de Janeiro, Brazil
[5]Laboratory of Respiratory Viruses and Measles, Fundacao Oswaldo Cruz, Rio de Janeiro, Brazil
[6]International Public Health, LSHTM, London, UK
[7]Clinical Research Department, London School of Hygiene & Tropical Medicine, London, UK
[8]School of Tropical Medicine and Global Health, Nagasaki University, Nagasaki, Japan
[9]Laboratório de Vírus Respiratórios e do Sarampo, IOC, Fundacao Oswaldo Cruz, Rio de Janeiro, Brazil

**Acknowledgements** Thanks are due to the families in Manguinhos who kindly agreed to participate in the study. This research would not have been possible without the support of Carlos Alberto de Moraes Costa, Iris Maria da Silva Lordello, and the staff of the Germano Sinval Faria Health Centre, the Oswaldo Cruz Foundation's COVID-19 Diagnosis Support Unit, Vice-Presidency of Health Production and Innovation and LabiExames.

**Contributors** PB: conceptualisation, methodology, writing, guarantor—original draft, writing—review and editing, project administration, and supervision. LD: data curation and methodology. TF: methodology, writing—original draft, writing—review and editing. LSB: visualisation and data curation. OGC: methodology, visualisation and data curation. FM: software and data curation. GAC: data curation. PR: resources. JW: funding acquisition and methodology. CS: funding acquisition and methodology. MMS: resources, MC: conceptualisation, methodology, writing—original draft, writing—review and editing.

**Funding** Ministério da Ciência, Tecnologia e Inovação Conselho Nacional de Desenvolvimento Científico e Tecnológico 3041101/2017-6, 307450/2021-0, 307282/2017-1, Fundação Carlos Chagas Filho de Amparo à Pesquisa do Estado do Rio de Janeiro E-26/201.356/2014 E-26/202.862/2018, E-26/211.565/2019, E-26/210.149/2020; UK Research and Innovation MR/V033530/1.

**Competing interests** None declared.

**Patient and public involvement** Patients and/or the public were involved in the design, or conduct, or reporting, or dissemination plans of this research. Refer to the Methods section for further details.

**Patient consent for publication** Not applicable.

**Ethics approval** This study involves human participants and was approved by Brazilian National Ethics Committee (CONEP) 30639420.0.0000.5262. Participants gave informed consent to participate in the study before taking part.

**Provenance and peer review** Not commissioned; externally peer reviewed.

**Data availability statement** Data are available on reasonable request. While the data at the individual level are not openly available to preserve the anonymity of the study participants, researchers who are interested in potential collaboration should contact the corresponding author. The study steering committee will evaluate data requests in accordance with best practices in cohort studies to ensure that data sharing would preserve confidentiality and be compatible with the terms of the informed consent form used in this study.

**ORCID iDs**
Patrícia Brasil http://orcid.org/0000-0001-9555-7976
Guilherme Amaral Calvet http://orcid.org/0000-0002-3545-5238
Chris Smith http://orcid.org/0000-0001-9238-3202

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
