## [Reviewer comments · BMJ Open]

ARTICLE DETAILS

TITLE (PROVISIONAL)	Cohort-profile: Household transmission of SARS-CoV-2 in a low-resource community in Rio de Janeiro, Brazil
AUTHORS	Brasil, Patrícia; Damasceno, Luana; Fuller, Trevon; Bastos, Leonardo; Cruz, Oswaldo; Medeiros, Fernando; CALVET, GUILHERME; Resende, Paola; Whitworth, Jimmy; Smith, Chris; Siqueira, Marilda; Carvalho, Marilia

VERSION 1 – REVIEW

REVIEWER	Enkeleint Mechili Faculty of Public Health, University of Vlora, Vlora, Albania, Department of Healthcare
REVIEW RETURNED	30-Aug-2022

GENERAL COMMENTS	This article is very interesting and it aims to better understand the local transmission of SARS-COV-2 in low-resource communities. However, to my view, the paper needs a lot of work in order to be published in a scientific Journal. It is not well structured, the methods are not presented well and the results section needs to be enriched with more info. Additionally, a discussion part is missing. Below you can find some comments: Abstract: - Please rephrase the purpose. In the sentence there are mentioned two different things;- Rows 16-23 should be removed and replaces with information about the methodology;- Rephrase row 38;- The results part is very unclear to me. Please add info;- Keywords in page 1 are not the same with those on page 5- The abstract should be re-written by providing more info on methods and results especially Introduction: - Row 11: mention the date because the number of cases and deaths change daily;- Rows 11-19 could be presented in a more formal way;- Row 6 a reference is needed;- The last paragraph of the introduction need to be transferred to methods (except the aim)- In general, the introduction is very complicated. It looks that jumps from one thing to another with no good connection. To my view it should be re-written. Cohort description - From row 30 to 47 no reference is available. Please add;- "Children who were brought to the primary local health centre for
---

	immunization, check-ups or emergency care were invited to participate” please specify that parents that brought children were invited ...  - This section should be structured. It looks that you have done a good job but it is not presented in an appropriate way; - As the study started in May 2020 it’s unclear to me this: “Vaccination status: number of doses, manufacturer, and vaccination date.” (row 33 page 6) - Which were the inclusion/exclusion criteria? Findings to date  - The first two sentences of this part are not clear. What do you mean the parent was not present? Please clarify this part; - To my view its better to present the different waves with time period also and not just with the name of the variant; - Page 11, row 16 “As enrolment began in May 2022”. Please correct; - From the first paragraph is not clear to me how many children in total were enrolled (not families) and also why some didn’t participate or were lost for follow up? - “Across all recruitment periods, 45-54% of the participants were adults” From this is not clear to me if you enrolled children only or adults? If you included adults this is not in line with what is presented above and with the title and aim of the study - No discussion is presented
--	--

REVIEWER	Adrian Martineau Queen Mary University of London
REVIEW RETURNED	24-Sep-2022

GENERAL COMMENTS	This manuscript profiles a SARS-CoV-2 cohort comprising children and adults from 842 households in Rio de Janeiro, Brasil. Analysis of the cohort has led to two publications to date.  1. Table 1 presents very limited information on participant characteristics – basically age and sex. Can more information be provided – ethnicity, socioeconomic status, educational level, household income, BMI, housing, smoking, vaccination status, type of vaccine administered etc? Clearly the authors have this data as per text on p9/22. 2. How do characteristics of cohort participants compare to the general population? Please provide a Table. This information is needed to inform issue of generalisability. 3. How do characteristics of participants who remained in follow-up differ from those lost to follow-up? Please provide a Table. 4. Data availability: supply of individual participant data need not compromise anonymity, as seems to be implied? Please edit / clarify 5. Can you clarify current recruitment status? This was not clear to me from the text. 6. Abstract: ‘The seroprevalence of SARS-CoV-2 was approximately 20% from May 2020 to June 2021 but increased to more than 80% for most age groups with the rollout of vaccination’. Does this refer to anti-S or anti-N antibody? Need to specify if this statement is to be meaningful and avoid confusion. The un-numbered figure on p22 also needs to specify what antibody is referred to.
---

VERSION 1 – AUTHOR RESPONSE

Reviewer: 1

Dr. Enkeleint Mechili, Faculty of Public Health, University of Vlora, Vlora, Albania, University of Crete School of Medicine

Comments to the Author:

This article is very interesting and it aims to better understand the local transmission of SARS-COV-2 in low-resource communities. However, to my view, the paper needs a lot of work in order to be published in a scientific Journal. It is not well structured, the methods are not presented well and the results section needs to be enriched with more info. Additionally, a discussion part is missing. Below you can find some comments:

Abstract:

Please rephrase the purpose. In the sentence there are mentioned two different things;

Replaced: "Purpose: To better understand the local transmission of SARS-COV-2 in low-resource communities and the role of children in such transmission, we have followed a cohort of children and their household contacts in Rio de Janeiro during the COVID-19 pandemic (2020 to 2022)."

With: "Purpose: To better understand the household transmission of SARS-COV-2 in a low-resource community in Rio de Janeiro during the COVID-19 pandemic (2020 to 2022)."

Rows 16-23 should be removed and replaces with information about the methodology;

Replaced: "We enrolled 842 families from May 20, 2020 to May 31, 2022. This period encompassed different locally-circulating SARS-CoV-2 lineages, in particular, B.1.1.33 during the first wave followed by variants Zeta (P.2), Gamma (P.1/P.1.*), Delta (B.1.617.2/AY.*), and Omicron (BA.*). The average duration of follow-up per family was 250 days."

With: "During home visits over 24 months we collected data on sociodemographic characteristics, behavioural data, clinical manifestations of SARS-CoV-2, vaccination status, SARS-CoV-2 RT-PCR and anti-S antibody tests"

Rephrase row 38;

Replaced: "household crowding was significantly associated with SARS-CoV-2 incidence."

With: "the risk of SARS-CoV-2 occurrence was higher in households with a high number of persons per room"

The results part is very unclear to me. Please add info;

Replaced: "The seroprevalence of SARS-CoV-2 was approximately 20% from May 2020 to June 2021 but increased to more than 80% for most age groups with the rollout of vaccination. During the first wave, children were not the principal source of SARS-CoV-2 infections in their households and household crowding was significantly associated with SARS-CoV-2 incidence. Through the Delta wave, the incidence among children was lower than that of older participants. Subsequently, vaccination of the elderly reached 90% and incidence among children was higher."

With: “We enrolled 845 families from May 2020 to May 2022. The median number of residents per household was four. The median household density, defined as the number of persons per room, was 0.95. The risk of SARS-CoV-2 occurrence was higher in households with a high number of persons per room. Children were not the principal source of SARS-CoV-2 infections in their households during the first wave of the pandemic.”

Keywords in page 1 are not the same with those on page 5

We corrected this.

The abstract should be re-written by providing more info on methods and results especially

We rewrote the abstract as described above.

Introduction:

Row 11: mention the date because the number of cases and deaths change daily;

We added the month and year in which we tabulated the number of cases and deaths.

Rows 11-19 could be presented in a more formal way;

Row 6 a reference is needed;

We have rewritten the Introduction as described below.

The last paragraph of the introduction need to be transferred to methods (except the aim)

We removed the sentence “We have tested the cohort participants at regular intervals for SARS-CoV-2 via RT-PCR of nasopharyngeal swabs and anti-spike IgG antibody tests of serum samples.”, as it is described in more detail in the **Cohort description** section.

In general, the introduction is very complicated. It looks that jumps from one thing to another with no good connection. To my view it should be re-written.

We have rewritten the Introduction to explain how the study fills a gap in our knowledge about household transmission of SARS-CoV-2 in low resource communities.

Cohort description

From row 30 to 47 no reference is available.

We added a reference to Carvalho et al. (2022)¹.

“Children who were brought to the primary local health center for immunization, check-ups or emergency care were invited to participate” please specify that parents that brought children were invited ...

We changed the wording as suggested.

This section should be structured. It looks that you have done a good job but it is not presented in an appropriate way;

To improve the organization of this section we separated it into subsections: The Epidemic Context, Recruitment and Eligibility Criteria, Follow-up, and Data collected

As the study started in May 2020 it's unclear to me this: "Vaccination status: number of doses, manufacturer, and vaccination date." (row 33 page 6)

We clarified that this information was collected after vaccination began: "Vaccination status: number of doses, manufacturer, and vaccination date (after vaccination began in January 2021)."

Which were the inclusion/exclusion criteria?

We added a subsection about the study's Recruitment and Eligibility Criteria.

Findings to date

The first two sentences of this part are not clear. What do you mean the parent was not present? Please clarify this part;

We clarified that the parent was not present at the time of the study visit but did reside in the household with the child.

To my view it's better to present the different waves with time period also and not just with the name of the variant;

We added the dates and cited an appropriate reference.

Page 11, row 16 "As enrolment began in May 2022". Please correct;

We corrected this typo.

From the first paragraph is not clear to me how many children in total were enrolled (not families) and also why some didn't participate or were lost for follow up?

We added: "This included 1152 children 12 and under, 758 of whom were retained in follow-up. As quarantine measures were relaxed, parents returned to working outside the home, which led to loss to follow-up of adults. This also resulted in loss to follow-up of children because children could not receive the study team in their homes without an adult present". Furthermore, in the Recruitment and Eligibility Criteria section we explain that individuals with only one contact were considered lost to follow-up and added tables comparing participants who were lost to follow-up to those who were retained in follow-up.

“Across all recruitment periods, 45-54% of the participants were adults” From this is not clear to me if you enrolled children only or adults? If you included adults this is not in line with what is presented above and with the title and aim of the study

We did enrol all adults who were the household contacts of eligible children. Had we only investigated the children living in a home, we would have obtained a very incomplete picture of the dynamics of transmission between family members of different ages. We added household transmission to the title.

No discussion is presented

This manuscript category does not have a Discussion.

Reviewer: 2

Dr. Adrian Martineau, Queen Mary University of London

Comments to the Author:

This manuscript profiles a SARS-CoV-2 cohort comprising children and adults from 842 households in Rio de Janeiro, Brasil. Analysis of the cohort has led to two publications to date.

1. Table 1 presents very limited information on participant characteristics – basically age and sex. Can more information be provided – ethnicity, socioeconomic status, educational level, household income, BMI, housing, smoking, vaccination status, type of vaccine administered etc? Clearly the authors have this data as per text on p9/22.

We have included the following variables: race (self-defined), educational level (completion of primary school), and smoking (Tables 1-2). Vaccine status is presented in Figure 4 illustrating changes in immunization rates over time. However, the description of the type of vaccine administered is complex, and is beyond the scope of the present analysis due to the use of heterologous vaccination schemes in Brazil. Education level is a better proxy for socio-economic status than household income hence we only included the former.

2. How do characteristics of cohort participants compare to the general population? Please provide a Table. This information is needed to inform issue of generalisability.

Manguinhos is similar to any “favela” in Rio de Janeiro² and probably to deprived communities in Latin America. Our study recruited the household contacts of children receiving care at a primary public health clinic. Based on this design, the cohort may not be generalizable to the general population. For example, the proportion of household members under 18 years of age is larger in our cohort than in the general population of Rio de Janeiro. By the same token, since we only recruited households with children, one cannot generalize from this cohort to households without children. The aim of cohort can be considered to gain insight into the process that gives rise to an outcome of interest in a particular population by studying the group over time. While the generalizability of our cohort may be limited, we believe that our cohort nevertheless provides valuable insights about disease transmission in this community such as elucidating the manner in which household density influences SARS-CoV-2 incidence.

3. How do characteristics of participants who remained in follow-up differ from those lost to follow-up? Please provide a Table.

These data are reported in the new Table 1.

4. Data availability: supply of individual participant data need not compromise anonymity, as seems to be implied? Please edit / clarify

As the area is relatively small, there could be a loss of anonymity if the data were freely accessible. For example, community health workers could identify the participants. To preserve the anonymity of the participants, in published maps of the study area we have added a random jitter to the coordinates of participants' households.

5. Can you clarify current recruitment status? This was not clear to me from the text.

We added: "Recruitment was closed at the end of this analysis, but it is currently reopened if new viral lineage is detected in the community."

6. Abstract: 'The seroprevalence of SARS-CoV-2 was approximately 20% from May 2020 to June 2021 but increased to more than 80% for most age groups with the rollout of vaccination'. Does this refer to anti-S or anti-N antibody? Need to specify if this statement is to be meaningful and avoid confusion. The un-numbered figure on p22 also needs to specify what antibody is referred to.

We updated the Abstract and Figure 3 caption to explain that it is anti-S antibody.